# Nanomedicine in Clinical Photodynamic Therapy for the Treatment of Brain Tumors

**DOI:** 10.3390/biomedicines10010096

**Published:** 2022-01-03

**Authors:** Hyung Shik Kim, Dong Yun Lee

**Affiliations:** 1Department of Bioengineering, College of Engineering, Hanyang University, Seoul 04763, Korea; kimhs5774@naver.com; 2Institute of Nano Science and Technology (INST), Hanyang University, Seoul 04763, Korea

**Keywords:** glioblastoma multiform (GBM), photodynamic therapy (PDT), photosensitizer (PS), reactive oxygen species (ROS), surgical resection, radiotherapy, chemotherapy, tumor microenvironment, blood–brain barrier (BBB), targeted therapy

## Abstract

The current treatment for malignant brain tumors includes surgical resection, radiotherapy, and chemotherapy. Nevertheless, the survival rate for patients with glioblastoma multiforme (GBM) with a high grade of malignancy is less than one year. From a clinical point of view, effective treatment of GBM is limited by several challenges. First, the anatomical complexity of the brain influences the extent of resection because a fine balance must be struck between maximal removal of malignant tissue and minimal surgical risk. Second, the central nervous system has a distinct microenvironment that is protected by the blood–brain barrier, restricting systemically delivered drugs from accessing the brain. Additionally, GBM is characterized by high intra-tumor and inter-tumor heterogeneity at cellular and histological levels. This peculiarity of GBM-constituent tissues induces different responses to therapeutic agents, leading to failure of targeted therapies. Unlike surgical resection and radiotherapy, photodynamic therapy (PDT) can treat micro-invasive areas while protecting sensitive brain regions. PDT involves photoactivation of photosensitizers (PSs) that are selectively incorporated into tumor cells. Photo-irradiation activates the PS by transfer of energy, resulting in production of reactive oxygen species to induce cell death. Clinical outcomes of PDT-treated GBM can be advanced in terms of nanomedicine. This review discusses clinical PDT applications of nanomedicine for the treatment of GBM.

## 1. Introduction

As photodynamic therapy (PDT) has been developed since the 1980s, many other treatment options have been improved. PDT has proven useful for many types of tumors, such as melanoma [1,2], esophageal cancer [3,4], and multidrug-resistant lung and breast cancers [5,6]. After other oncology applications, interest in PDT as a high-grade glioma treatment stems from both the nature of tumor growth and the limited effectiveness of modern therapies available to this patient population [7]. Although surgical resection, partial radiation, and chemotherapy are key treatments for intracranial brain tumors, the invasive growth patterns, particularly in the central region of the cerebrum, complicate total resection. Unlike surgical resection and radiation, PDT can treat micro-invasive areas while protecting sensitive brain areas. These advantages over conventional therapies have been reported to improve outcomes in patient populations with overall very poor survival and incidence of iatrogenic injury.

Clinical trials for brain tumors exist to date, but there are many questions about PDT and its usefulness as a standard adjuvant therapy. First, the effect of PDT alone reported in clinical trials were positive based on parallel administration of standard treatment. Variables that should be standardized across further studies include photosensitizer (PS) selection, injected dose, irradiation light wavelength, sensitivity of brain tumor types, and adjuvant use of chemotherapy and radiation. Furthermore, additional studies are needed to enhance the targeting of brain tumors while considering the pharmacokinetic aspects and methods of improving the quantum yield of the PS, which generates effective reactive oxygen species under light irradiation. In this regard, the rapidly developing fields of nanotechnology and nanomedicine are producing nanostructured materials that can overcome the shortcomings of delivery systems used in clinical practice. In fact, the functional presence of the blood–brain barrier (BBB) limits the delivery of drugs to the brain tumors. To overcome this limitation, many strategies for temporarily opening the BBB through physical impact such as magnetic resonance (MR)-guided focused ultrasound have been studied recently, but this raises a problem in compatibility [8]. Therefore, the use of multifunctional nanocarriers as drug delivery systems is emerging as one of the most promising strategies [9]. In general, intracellular transport of nanocarriers is mediated by the vesicular system, and three types of intracellular vesicles are involved (e.g., clathrin-mediated, caveolae-mediated and macropinocytotic vesicles) [10]. Therefore, in order to pass through these pathways, nanocarriers covalently bound with specific targeting ligands to guide the drug across the BBB to specific sites in each tumor type. The physicochemical and mechanical properties of nanocarriers differ depending on the material, size, shape (mesoporous structure, rod shape, particle), and the selected ligand. This allows customization for increased brain-targeted delivery of PS or therapeutic drugs. Although many PS nanocarriers are still in the early stages of translation, many advances have been made in recent years for functional nanomedicines based on BBB crossing.

Another advantage of nanoparticles is that they can increase the low solubility of PS, prolong blood circulation, promote targeted delivery and cellular uptake, while protecting the drug from degradation. This makes it an interesting alternative to traditional PDT because the nanostructures can enable efficient transport of PS and ameliorate the lack of anticancer activity [11]. To date, in addition to micellar self-assembly techniques for PS delivery, numerous nanoparticles, such as gold, silica, upconversion, and carbon-based particles, have been studied to increase their phototoxic properties and to increase their concentrations in tumor sites.

In this study, we classify brain tumors according to malignancy and examine the applicability of PDT for each type. We discuss the use of PDT and the properties and clinical applications of nanoparticles as potential delivery tools for PS delivery. In addition, the possibility of application to brain tumors is discussed through clinical cases of nanomedicine-based PDT.

## 2. Classification of Brain Tumor Grade

A brain tumor, a tumor that develops within the skull, is an abnormal mass of tissue in which cells grow and multiply out of control. Although more than 150 types of brain tumors have been reported, they are macroscopically divided into primary and metastatic groups [12]. Tumors that arise directly from the brain tissue or surrounding the brain are classified as primary brain tumors. More specifically, they are classified as either glial composed of glial cells or comparative cells that arise from brain structures containing blood vessels, nerves, and sweat glands [13]. Metastatic brain tumors, commonly considered malignant tumors, include tumors that develop elsewhere in the body, such as the lungs or breast, and travel through the bloodstream to the brain It is reported that there are more than 150,000 tumors that have metastasized to the brain each year, accounting for about 25% of cancer patients [14]. Typically, up to 40% of lung cancer patients develop metastatic brain tumors, and the survival rate of those diagnosed with this tumor is very fatal, typically taking only a few weeks from diagnosis to death. A grading system was developed by the World Health Organization (WHO) to indicate whether a tumor is malignant or benign based on histological features observed under the microscope, such as most malignant, widely invasive, rapidly growing and prone to aggressive necrosis, and rapid recurrence (Table 1) [15].

### 2.1. Types of Low Grade (Grade I and Grade II) Brain Tumors

#### 2.1.1. Craniopharyngiomas

Craniopharyngiomas are a rare type of benign brain tumor and do not spread to other tissues, but they are difficult to remove because they are located deep in the brain and near critical structures such as the pituitary gland. They usually affect the function of the pituitary gland, which regulates many hormones in the body; thus, almost all patients are treated with hormone replacement therapy.

#### 2.1.2. Chordomas

Chordomas are rare axial skeletal malignancies that most commonly occur in people between the ages of 50 and 60, accounting for less than 1% of intracranial tumors and 4% of bone tumors [16]. Because the most common locations are below the spine and at the base of the skull, they can invade adjacent bones and put lasting pressure on the surrounding nerve tissue [17]. Radiation therapy is the most practiced treatment, but radiation dose is limited because stability of important nerve structures such as the brainstem and nerves must be ensured. Therefore, highly focused radiation therapy such as carbon ion therapy and proton therapy are known as a more effective treatment method than conventional X-ray radiation [18].

#### 2.1.3. Gangliogliomas and Gangliocytomas

Gangliogliomas, gangliocytomas, and anaplastic gliomas are rare tumors containing glial cells and relatively well-differentiated neoplastic nerve cells [19]. They usually develop in the two temporal lobes, one on each side of the brain around the ear. In this location, these tumors tend to cause epilepsy; thus, seizures can be the first sign of ganglioglioma [20]. Gangliogliomas are rare, occur primarily in young adults, and account for about 1 to 2% of all brain tumors.

#### 2.1.4. Schwannomas

Schwannoma is a benign brain tumor that usually arises from cells involved in electrical insulation of nerve cells, and the diagnoses are high in adults between the ages of 20 and 50 years [21]. Schwannomas are also called neuromas, neurolemomas, or neurilemomas. A typical schwannoma is an auditory neuroma, which arises from the eighth cranial nerve or vestibular cochlear nerve, which lines the brain from the ear. Because schwannoma is found in the outer skin surrounding the nerve, radiation surgery is widely used as a treatment and surgery can be completed without nerve damage, except for vestibular schwannoma, which frequently causes hearing loss [22].

#### 2.1.5. Pituitary Adenomas and Pineocytomas

Pituitary adenoma is the second most common intracranial tumor after glioma, meningioma, and schwannoma. Most pituitary adenomas are benign and grow slowly. Adenoma is the most common disease affecting the pituitary gland, and even malignant pituitary tumors rarely spread to other parts of the body [23]. Pineocytoma is a benign lesion that usually develops in the cells of the pineal gland and mainly occurs in adults [24]. They are mostly homogeneous, non-invasive, and slow growing. Most of these tumors can be successfully treated.

### 2.2. Types of High Grade (Grade III and Grade IV) Brain Tumors

Glioma is the most common type of adult brain tumor, accounting for 78% of malignant brain tumors [25]. They occur in supporting glial cells in the brain which are subdivided into astrocytes, oligodendrocytes, and ependymal cells. These glial tumors are discussed in the following sections [26].

#### 2.2.1. Anaplastic Astrocytomas

Although astrocytoma can occur in many parts of the brain, it most commonly occurs in the cerebrum and is the most common type of glioma, accounting for more than half of all primary brain and spinal cord tumors [27,28]. Anaplastic astrocytoma is considered a more malignant evolutionary form of the previously lower-grade astrocytoma, with more aggressive features, including a faster growth rate and greater invasion into the brain. Histologically, it shows greater cellular abnormalities and evidence of cell proliferation (mitosis) compared to Grade II tumors. Surgical resection is not considered a complete cure for these tumors and will always need to be followed by radiation therapy and chemotherapy [29].

#### 2.2.2. Anaplastic Oligodendrogliomas

Oligodendrogliomas are generally found in the white matter and the outer layer of the brain called the cortex but can arise anywhere in the central nervous system [30]. They are usually derived from cells that produce myelin, the insulator for nerves in the brain. Oligodendrocytes are classified into two grades, Grade II and Grade III, based on growth rate and invasiveness; the more malignant ones are called anaplastic oligodendrogliomas. The first line of treatment for anaplastic oligodendrogliomas is surgical resection, if possible. The goal of surgery is to excise tissue to determine the type of tumor and to remove as many tumors as possible without causing more symptoms in the patient [31]. Treatment after surgery can include radiation, chemotherapy, or clinical trials.

#### 2.2.3. Glioblastoma Multiforme (GBM)

Glioblastoma multiforme (GBM) consists of several types of cells, such as astrocytes and oligodendrocytes, which are the most aggressive, malignant and common forms of neuronal supporting astrocytoma that develop within the brain. It is characterized by cells that appear histologically abnormal, proliferation, areas of dead tissue, and formation of new blood vessels [32]. GBM presents as a malignant progression in a previously present lower-grade astrocytoma in less than 10% of cases. In more than 90% of cases, it can begin as a Grade IV tumor. The standard approach to treatment in the newly diagnosed setting includes postoperative concomitant radiation therapy with temozolomide and additional adjuvant temozolomide. Unfortunately, there is no standard treatment for relapses. However, surgery, radiation therapy and chemotherapy or systemic therapy with bevacizumab are all options depending on patient circumstances [33].

## 3. Strategies to Improve Permeability of Nanocarrier through the Blood-Brain Barrier

One of the major limitations of treating brain tumors is the difficulty of delivering drugs to the brain. The brain is surrounded by the blood–brain barrier (BBB), a selective barrier formed by endothelial cells in the cerebral microvessels, which regulates nutrient and ion transport and protects the brain from neurotoxic molecules to maintain brain homeostasis [34]. Unfortunately, most drugs cannot cross the BBB via physiological pathways due to the extreme selectivity of the barrier, which constitutes the greatest obstacle to systemic treatment for most central nervous system (CNS) diseases. In the recent decade, many strategies have been studied, such as topical delivery, implantation of a sustained drug-release scaffold [35], nasal administration [36], ultrasound to temporarily open the BBB [37], and nanoparticle functionalization to enhance BBB penetration [38]. However, local drug delivery methods are considered highly invasive because they require procedural surgery. In addition, the intranasal route has a disadvantage in that the delivered dose varies greatly depending on the condition of the nasal mucosa. Therefore, despite the difficulties across the BBB, the most popular and well-studied delivery route remains the systemic route through the functionalization of nanoparticles.

Nanocarriers can traverse the BBB using a variety of physiological pathways, including receptor-mediated transcytosis (RMT) or adsorption-mediated transcytosis (AMT). To achieve this goal, many nanocarrier systems, such as inorganic, polymeric, or lipid-based nanoparticles, have been developed and shown to cross the BBB due to their tailored surface properties. Numerous studies have demonstrated that physically coating nanoparticles with surfactants and chemical functionalization with specific ligands is a successful strategy to enhance BBB traversing via the physiological pathways mentioned above [39,40]. The size and charge of nanoparticles are also aspects that can affect brain penetration, but if the surface functionalization is done properly, there is no significant difference in a wide size range (from 5 to 400 nm) [41]. Smaller nanoparticles can cross the BBB more easily and diffuse better through the brain, but larger nanoparticles can also cross the BBB in slightly smaller amounts when properly functionalized. On the other hand, larger particles can load a greater amount of drug but reach the brain at a lower concentration, and smaller nanoparticles cannot contain a large amount of drug but reach the brain at a higher concentration. Therefore, the key to increasing the amount of drug delivered to the brain is finding the optimal particle size and designing a nanoparticle system that fits the purpose.

## 4. Advantages and Clinical Application of PDT for the Treatment of Brain Tumors

### 4.1. PDT Mechanism and Advantages for Brain Tumor Treatment

PDT is a therapy in which treatment is implemented through photoactivation of PS present or selectively accumulated around tumor cells. Photoirradiation activates PS by energy transfer of PS with structural specificity that excites molecular oxygen to singlet or triplet states. In the singlet state, the first step through excitation, energy is converted to heat by internal conversion or emitted as fluorescence. In the triplet state of a stage capable of reacting with surrounding or tissue oxygen, energy induces cell death by generating reactive oxygen species (ROS) (Figure 1). The generated ROS reacts rapidly with macromolecules that make up cells, including unsaturated fatty acids, proteins, and cholesterol, and this reaction destroys the membranes of intracellular organelles, such as lysosomes, mitochondria, and endoplasmic reticulum, which are directly related to cell viability [42]. Therefore, PDT ultimately induces apoptosis and necrosis of tumor cells, as well as inhibits tumor cell growth by generating local ischemia due to occlusion of tumor microvessels that supply nutrients and oxygen to the tumor. Furthermore, damage-associated molecular patterns (DAMPs) and various cytokines secreted by tumor cell death activate the subsequent host immune response, providing an opportunity for combination with immunotherapy [43]. Therefore, cancer treatment through surgery, chemotherapy, radiation, immunotherapy, monoclonal antibody, and various combinations thereof are currently being applied, and the selection of each combination is determined by considering the type and stage of the disease and the patient’s overall health.

A major problem with most existing cancer therapies, such as chemotherapy and radiation, is the combination of high toxicity to the patient’s nonspecific cells and low specificity to cancer cells [44]. PDT is a promising alternative method for treating brain tumors, as PSs are used to target specific dysfunctional cells, and light is targeted to specific location to induce destruction of oncogenic cells [45]. Moreover, PS photodynamic activity results in release of cytotoxic ROS, such as single-molecule oxygen, based on photo-oxidation reactions that trigger many subsequent biochemical and molecular reactions [46]. Unlike surgical resection and radiotherapy, PDT can treat micro-invasive areas while protecting sensitive brain regions [47]. These advantages over current therapies can reduce the incidence of iatrogenic injury and improve outcomes in patient populations with poor survival and recurrence rates from GBM.

### 4.2. Clinical Trials of PDT for Brain Tumors

The therapeutic application of PDT for cancer began in the late 1970s with tests of the effects of light irradiation on hematoporphyrin derivatives (HPDs) in five patients with bladder cancer [48]. Since this initial work, more than 250 clinical trials of PDT have been conducted, with targets ranging from pre-malignant skin cancers [49,50,51,52,53,54] to peritoneal carcinomatosis [55,56], gastrointestinal cancers [57,58,59,60,61,62,63,64], lung cancers [65], and brain tumors [66,67]. Although many clinical trials of PDT for the treatment of malignant brain tumors have been conducted, most of them are phase I or II trials [66]. The heterogeneity of adjuvant therapies and tumor subtype therapies in the procedures used in clinical studies of the effects of PDT on brain tumors hampered the evaluation of the effectiveness of PDT.

In the 1990s, the PDT with PHOTOFRIN^®^ (porfimer sodium) was evaluated at low or moderate light intensity, and in some cases, intraoperative adjuvant therapy was applied [68,69]. The above clinical trial was conducted on newly diagnosed GBM patients and recurrent GBM patients, and the results were significant. They had overall survival (OS) values of 6–9 months for newly diagnosed GBM [70] and 6–7 months for recurrent GBM [71]. Furthermore, in 2006, the researchers conducted a large phase III clinical trial (NCT00003788) with subjects with 150 newly diagnosed patients and 120 patients with recurrent glioma. At this time, the concentration of the photosensitizer was fixed at 2 mg/kg of PHOTOFRIN^®^, but the amount of light was varied with an average of 58 ± 17 J/cm^2^. However, the results showed that the OS of newly diagnosed GBM was 7.6 months and the OS of recurrent GBM was 6.7 months, which was not improved compared to previous studies. Furthermore, a phase 1 clinical trial (NCT01682746) was initiated using step-by-step dose escalation of PHOTOFRIN^®^ and fixed light intensity to determine the maximum safe dose for pediatric patients. Three patients with infratentorial tumors were enrolled and treated with PHOTOFRIN^®^ at 0.5 mg/kg, and PDT did not adversely affect these patients.

In addition, PDT was applied in light dose escalation studies up to 230 J/cm^2^ in phase I or II trials, resulting in OS values of 14.3 months for newly diagnosed GBM and 13.5 months for recurrent GBM [72,73]. The team conducting the clinical trial proceeded to a therapeutic condition in which light could penetrate deep enough into solid tumors to reach and kill migrating cancer cells without affecting normal cells. Therapeutic application was achieved by an i/o plane-cut laser fiber inserted into the lipid pool of the resection cavity, and the results were superior to those of other examinations at the time and comparable to the current standard of care. A phase II clinical adult trial (NCT01966809) was conducted using these conditions, but subsequent results were not reported because it aimed to reproduce the reported survival improvement and to define the antitumor activity of PHOTOFRIN^®^.

A total of 136 patients, including 78 patients with GBM and 58 patients with anaplastic astrocytoma who underwent tumor resection in 2005, was treated with an HPD at 5 mg/kg and then laser irradiated [73]. For newly diagnosed patients, 73% and 25% of patients with anaplastic astrocytoma and GBM survived at least 36 months, respectively, and 57% and 41% of patients with anaplastic astrocytoma and GBM, respectively, survived after repeated surgery.

Large phase III clinical trials aimed to study the efficacy and safety of 5-aminolevulinic acid (5-ALA) in combination with PDT in patients with high-grade glioma (HGG) [74,75]. 5-ALA and PHOTOFRIN^®^ PDT, irradiated at a wavelength of 630 nm, using an implanted catheter, were performed on primary GBM patients on the day of fluorescence-guided surgery (FGS) after the patient recovered from surgery. Patients in the control group underwent traditional surgical resection, whereas patients who received PHOTOFRIN^®^ PDT therapy received a total of 5 PDT sessions at daily intervals. Compared with conventional surgery, the mean tumor progression was delayed by 3.8 months in GBM patients treated with PDT after FGS, and the mean survival increased from 24.6 weeks to 52.8 months. 5-ALA has been reported to have fewer side effects when applied to HGG compared to PHOTOFRIN^®^ PDT. PHOTOFRIN^®^ has been reported to damage normal brain tissue due to vascular occlusion of the compound and concluded that it increases the risk of nerve damage and permanent defects at total applied light doses greater than 4000 J using diffuse tip fibers [76]. On the other hand, 5-ALA could be safely applied to patients in the high energy range of 4320 to 11,520 J. It was also noncytotoxic when applied systemically and did not appear to be significantly redistributed by edema volume flow around the tumor.

## 5. Nanotechnology for Enhanced Photodynamic Therapy

Nanotechnology generally involves development of materials with dimensions between 1 and 100 nm, a scale at which the properties of materials differ significantly from those of bulk materials and can be tailored to the desired application [77]. These new chemical and physical properties are usually derived from rapidly increasing surface-to-volume ratios and are associated with plasmonic and quantum effects. With rapidly advancing nanotechnology, nanomaterials are excellent therapeutic and diagnostic tools, and thousands of new compounds and nanostructures are developed each year for diverse applications [78]. This approach using nanotechnology could help overcome several obstacles that have prevented photodynamic therapy from attaining widespread clinical success. The nanostructures studied so far have been applied as a drug delivery platform for PDT and as a strategy to improve the efficiency of photosensitizers that generate ROS upon irradiation. Nanoparticles can be made up of a variety of components, organic and inorganic; can have a variety of shapes and sizes within the nanoscale scale and can act as photosensitizers or as energy converters [79]. Moreover, nanocarriers prevent aggregation caused by the low solubility of photosensitizers in aqueous media such as blood and bypass healthy tissues to increase tumor accumulation. In this section, we discuss cases where nanocarriers provide sufficient therapeutic efficacy and address issues such as undesirable biodistribution or rapid drug clearance from tumor areas.

### 5.1. Recent Advances in Preclinical Application of Nanocarriers for PDT

Although modern PDT has significantly improved the quality of life and increased overall survival of cancer patients, it is important to further improve the therapeutic effects of nanocarriers to minimize notable side effects such as hydrophobic PS and off-target side effects. In this regard, researchers have studied numerous nanocarriers such as polymers, liposomes, micelles, inorganic oxide, and novel metal nanoparticles to increase the therapeutic efficacy of photosensitizers (Table 2). First and foremost, it is important to utilize nanocarriers to efficiently deliver photosensitizers and generated singlet oxygen molecules to the target site in an optimal therapeutic range. The pharmacokinetic or pharmacodynamic characteristics of the nanocarriers should be confirmed for clinical use. Therefore, these are being studied for diagnostic as well as photodynamic/chemotherapeutic application using multifunctional nanoparticles.

### 5.2. Self-Assembled NP via Transformation into Amphiphilic PS-Derivatives

Second-generation PSs have a variety of functional groups including carboxyl, hydroxyl and amine groups in addition to their basic porphyrin structure, allowing hydrophobic modifications via chemical or physical approaches to form amphiphilic PS derivatives. The PS derivatives synthesized to be amphiphilic can form NPs with various nanostructures such as micelles [89,95,96,97,98], PS-drug conjugates [82,86,92,99,100], polymersomes [101], and nanogels [84,102] through self-assembly. According to the nanostructure, nanoparticles can be generally categorized into three types: 1) NPs with mixed hydrophilic and hydrophobic domains, 2) NPs with a core-shell structure, and 3) NPs with a double-layered capsule structure (Figure 2). In this respect, natural polysaccharides such as hyaluronic acid (HA) [102,103,104,105], chitosan [102,106,107,108], chitin, heparin [95,99,100], and fucoidan [84] have been utilized as potential photosensitizer carriers due to their biocompatibility and biodegradability. In addition, not only polymers but also hydrophobic small molecule anticancer drugs (such as doxorubicin [96], docetaxel [97,107,109,110], paclitaxel [85,93,103,111,112], camptothecin, and quercetin [113,114]) can be grafted onto PS.

#### 5.2.1. Self-Assembly Methods for Amphiphilic PS Derivatives

Until recently, various approaches have been tried and developed to promote the self-assembly behavior of amphiphilic PS derivatives. In general, the dispersion method is a suitable method for an amphiphilic material having high water solubility to prepare nanoparticles with a core-shell structure [115]. The process can include mechanical agitation, mild heating or sonication. As an alternative to amphiphilic PS derivatives with low solubility in aqueous media, dialysis has been the most reported method [116,117]. On this basis, the PS derivative can be dissolved in an organic solvent such as dimethyl sulfoxide, dimethyl formamide, or methanol mixed with water and dialyzed against an aqueous solution to remove the solvent. Alternatively, the emulsion method, in which the drug is encapsulated in the oil phase of an oil-in-water emulsion, has received much attention because the drug-loading system has a high loading efficiency [118]. In general, controlled drug release was achieved with the development of dual emulsion technology via water-in-oil in water emulsions [119]. For instance, the structural design of nanocarriers using polyethylene glycolated poly(lactide-co-glycolide) (PEG-PLGA) were obtained as the most suitable approach in nanoemulsions, such as a water-in-oil-in-water evaporation process. In the hydrophilic part of the nanocarrier, cisplatin, a cell proliferation inhibitor, was placed and encapsulated by placing the hydrophobic porphyrin photosensitive dye verteporfin in the oil phase. As a result, PLGA nanocarriers were enabled to efficiently deliver hybrid cargo to cancer cells and PDT-supported enhanced apoptosis.

#### 5.2.2. Carboxyl Group Modification of PS-Derivatives

The carboxyl modification of PS for linking to hyaluronic acid, chitosan, and heparin is mainly achieved through esterification or amidation. Esterification is used to join hydroxyl groups from PS to carboxyl groups in modifiers mediated by coupling agents or catalysts such as dicyclohexyl carbodiimide (DCC) and 4-dimethylaminopyridine (DMAP) (Figure 3a). Most carbodiimides are nonhydrophilic, limiting their application in hydrophilic systems of PS, with the exception of 1-(3-dimethylaminopropyl)-3-ethylcarbodiimide hydrochloride (EDC), which is extensively used in PS preparation. Another approach for modifying PS is to form an amide bond between the amine group of the hydrophobic material and the carboxyl group of the PS, where EDC and N-hydroxysuccinimide (NHS) are commonly used as condensing agents and catalysts, respectively (Figure 3a).

#### 5.2.3. Hydroxyl Group Modification of PS-Derivatives

In the case of hydroxyl groups, there are several representative reactions such as etherification and esterification that occur in the presence of alkylating or acylating agents (Figure 3b). As an example, light-controllable host–guest supramolecular amphiphilic complexes between azobenzene mediation porphyrin (TPP-Azo) were synthesized by esterification between TPPC6-COOH and AzoC6-OH [120]. Moreover, dextran alkyl carbonates were synthesized using various types of acylating agents such as butyl chloroformate, butyl fluoroformate, and ethyl chloroformate [121]. A cholesteryl hemisuccinate grafted hyaluronan synthesized through esterification of the carboxyl and hydroxyl groups of hemisuccinate-modified cholesteryl in the presence of DCC/DMAP has also been reported [81].

#### 5.2.4. Amine Group Modification of PS-Derivatives

Heparin [122] and carboxymethyl chitosan [123] of the adipic acid dihydrazide(ADH)-modified polysaccharide type were rendered hydrophobic directly by conjugation of the amino group with the carboxylate moiety of PS via an amino bond in the presence of a catalyst such as EDC/NHS. Another typical strategy is a condensation reaction where the amine group of PS reacts with a carbonyl compound to form an imine intermediate and then is reduced under NaBH_3_CN (Figure 3c). For these synthetic methods, the most important consideration is the determination of suitable solvents for both hydrophilic PS and hydrophobic molecules.

Chitosan has functional groups such as hydroxyl and amine; therefore, it can be easily modified and crosslinked with other polymers. Of particular benefit in terms of drug delivery, the amino groups of chitosan can be protonated in acidic environments, leading to pH-responsive behavior favored by acidic intracellular organelles such as endosomes and lysosomes. Therefore, chitosan-based nanoparticles have aroused great interest in the field of bio-nanomedicine, especially drug delivery. A dual reactive nanosystem comprised of indocyanine green (ICG) loaded mesoporous silica nanoparticles covered with ZnO quantum dots and coated with erlotinib-modified chitosan for synergistic photodynamic/molecular targeted therapy has been reported. The nanosystem showed a fairly distinct distribution in various nonsmall cell lung cancer models, with favorable anticancer results [124]. Moreover, biodegradable polymer nanoparticles based on chitosan that conjugate various amounts of the photosensitizer tetraphenylchlorin have been developed. These nanoparticles showed high drug loading efficiency and strong retention due to hydrophobic interactions such as π-π stacking between the aromatic photosensitizer group of the polymer and the drug. Nanoparticles have an excellent photodynamic therapeutic effect through photo-induced photochemical activation through high-dose drug delivery, and thus have a strong therapeutic effect on breast cancer cells [125].

#### 5.2.5. Hyaluronic Acid-Modified NPs for PDT

HA is rich in functional groups including carboxyl, hydroxyl and N-acetyl groups, is ready to be transformed into a hydrophobic material, and has a negative charge that can provide a binding platform for hydrophobic macromolecules with a positive charge. Notably, its bioactivity binding to receptors upregulated in cancer cells, such as the cluster determinant 44 (CD44) receptor, the HA-mediated motility receptor (RHAMM), and the lymphatic endothelial (LYVE)-1, allows it to be used for targeted therapeutics. Therefore, HA can act as both a carrier and a target receptor, and HA-based NPs have been extensively studied in the field of drug delivery. HA-related nanosystems (AuNCs-HA) for decorating gold nanocages were developed and exhibited significant photocatalytic properties for PDT, large surface areas, and photothermal therapy (PTT) or PDT properties under near-infrared (NIR) stimulation. In vivo assays showed complete inhibition through the combination of PDT and PTT in AuNCs-HA-treated tumor cells than when each therapy was treated individually [126]. In another study, 5-ALA, Cy7.5 and anti-HER2 antibodies were conjugated to HA and mounted on a gold nanorod (GNR) surface to yield multifunctional GNR-HAALA/Cy7.5-HER2 nanoplatform. As a result, the tumor targeting by HER2 was improved, and side effects were minimized, and the combination of PDT and PTT mediated by 5-ALA and Cy7.5 effectively caused tumor regression [127].

### 5.3. Application of Inorganic Nanomaterials in PDT

#### 5.3.1. Silica Nanoparticles

Although silica lacks PDT activity on its own, silica nanoparticles can be used to encapsulate PS in PDT due to the chemically inert, nontoxic, and optically transparent nature of silica [128]. In addition, it is commonly used for drug delivery in research because it is possible to functionalize chemicals to the silica through the hydroxyl groups on the silica surface (Figure 4) [129,130]. Mesoporous silica nanoparticles (MSNs) have been extensively utilized to deliver PSs, typically due to their interesting features such as large surface area and pore volume as well as high chemical stability [131,132,133]. One group has developed mesoporous silica-based nanoparticles to exploit continuous oxygen evolution to enhance the effectiveness of PDT treatment in hypoxic cancer environments [132]. To assemble Fe_3_O_4_ nanocrystals on silica nanoparticles doped with mesoporous dye, the surface was treated with 3-aminopropyltriethoxysilane and functionalized with amine groups. The oleic acid-stabilized Fe_3_O_4_ nanocrystals synthesized in an organic medium were reacted with the amine group of 2-bromo-2-methylpropionic acid, and the resulting Fe_3_O_4_ nanocrystals were assembled on the MSN surface by direct nucleophilic substitution between terminal bromine groups. The synthesized biocompatible manganese ferrite nanoparticle-immobilized mesoporous silica nanoparticles alleviated the hypoxic state of tumors with only a small number of nanoparticles and improved the treatment outcome of PDT in vivo.

NIR light-reactive multifunctional nanoparticles are ferrocene-modified with ICG rods and β-cyclodextrin (β-CD) capping for cooperative chemo-dynamic/photothermal/photodynamic (CDT/PTT/PDT) NPs made of mesoporous silica [131]. As a mechanism of chemo-dynamic therapy, ferrocene released from multifunctional nanoparticles was able to efficiently kill cancer cells by converting intracellular H_2_O_2_ into toxic OH through a ferrocene-mediated Fenton reaction. Moreover, ^1^O_2_ generated by ICG from near-infrared irradiation can kill cancer cells in cooperation with PDT. The results of in vitro experiments show that the CDT/PTT/PDT collaboration significantly amplified the inhibition rate of HeLa cells.

It was reported in one study that silica nanoparticles modified with folic acid (FA) could enhance the site-specific delivery of PS chlorin e6 (Ce6) [134]. By improving the efficiency of targeted drug delivery by FA, efficient generation of singlet oxygen at 670 nm irradiation was obtained, which improved the killing efficacy of NPs on MDA-MB-231 cells compared to free Ce6. Furthermore, a perfluoro hexane (PFH)-encapsulated MSN-based multifunctional nanoplatform using the PS ICG loaded into a polydopamine (PDA) layer and PEG-FA decoration was presented [135]. When excited with 808 nm light irradiation, it mediates the vaporization of PFH, creating bubbles for tumor ultrasound imaging and simultaneously inducing burst drug release. The PTT effect was exerted on the PDA layer, and the loaded ICG was able to generate ROS, a PDT mechanism, while providing NIR fluorescence emission.

#### 5.3.2. Gold Nanoparticles

Gold nanoparticles have been studied for many years for effective PDT induction as well as drug carriers due to promising properties such as high surface area, facile surface modification through gold thiol chemistry, and biocompatibility. Furthermore, gold nanoparticles are being extensively studied for diagnostic applications because of their ability to tune optical scattering and absorption via physical features such as surface plasmon resonance effects [136]. Gold nanoparticles can be applied to PDT without the use of an organic PS. The first use of gold nanorods (AuNRs) alone was reported in 2014 [137]. Upon excitation with relatively long-wavelength NIR light (915 nm), gold nanorods were able to generate a singlet oxygen (^1^O_2_) and destroy B16F0 melanoma tumors in mice. Excitation of gold nanorods at a wavelength of 780 nm (λ2), at which the PTT effect can be expected after generation of ^1^O_2_, increases the temperature around the tumor tissue, as confirmed by formation of heat shock protein (HSP 70) in which photon energy is converted into heat. By changing the activation wavelength band, the dominant phototherapeutic effect can be switched between PDT and PTT and a synergistic effect can be obtained. It was also possible to trace the distribution of gold nanorods in vivo through self-emitting single-photon-induced fluorescence.

The same group tested the effect of PDT by comparing different types of gold nanoshells, including nanorod-in-shell, nanocage and nanoparticle-in-shell, and demonstrated that it could completely eliminate solid tumors in mice [138]. They can modulate and switch the dominant roles of PDT and PTT by altering the activation wavelength that can excite the gold nanocage. As the most optimal conditions suggested by them, the nanocages mostly showed PDT effect when excited by 980 nm light, whereas 808 nm irradiation induced effective PTT. In vivo studies at 940 nm excitation, a wavelength band between 980 nm and 808 nm, demonstrate that gold nanoshells could induce dual-mode PDT/PTT for more efficient treatment of B16F0 melanoma tumors than that of doxorubicin, a clinically used drug.

Another group found that singlet oxygen could be produced when irradiated with a wide range of wavelengths (660–975 nm) [139]. Even under low-intensity light irradiation of 200 mW/cm^2^, the highest production of ^1^O_2_ was observed when a wavelength overlapped with the localized surface plasmon resonance (LSPR) peak, which is a characteristic of gold nanoparticles.

Many previous studies have demonstrated the ability of metal nanoparticles to efficiently excite PS through a single-photon excitation mechanism to generate singlet oxygen, which has been applied to typical PDT therapy [140,141]. However, one-photon excitation can cause potential photodamage to tissues adjacent to the tumor site due to the high energy provided by the comparatively short light wavelength. Therefore, two-photon excitation that precisely manipulates the therapeutic dose is preferable in this sense. To overcome this, a two-photon PDT was developed using a femtosecond laser beam capable of obtaining a high luminous flux. In one study, two-photon-induced singlet oxygen generation was observed by irradiating femtosecond laser pulses at 800 nm to aggregates of gold nanospheres and gold nanorods developed using non-agglomerated or aggregated gold nanoparticles [142]. As a result, the ^1^O_2_ generation capacity in gold nanoparticle was generally enhanced by the agglomerated state and was 8.3 times higher than that of the non-agglomerated gold nanoparticles. A similar trend was observed when the agglomerated gold nanorods were used; the singlet oxygen production efficiency was improved by 1.8 times compared to the non-agglomerated gold nanorods.

With the rapid advances in nanotechnology, there are a variety of synthetic methods available to researchers to obtain gold nanoparticles with suitable structures and features for PDT applications [143]. In addition to the various physicochemical properties, the additional chemical modification potential mentioned above could improve bioavailability and usability, suggesting gold nanoparticles as a promising candidate for clinical cancer treatment.

#### 5.3.3. Graphene Nanomaterials

Graphene-based nanomaterials, including graphene oxide (GO) and graphene quantum dots (GQD), have been widely used for cancer treatment such as anticancer drug delivery and PDT [144,145,146]. GO produced through oxidation process shows more favorable properties in terms of PS transport mediation due to improved water solubility and various functionalization chemistries. Characterized by abundant oxygen-containing moieties on their surface, GO nanomaterials allow further modification by many functional molecules such as targeting agents, activators and hydrophilic macromolecules, expanding biological applications and reducing toxicity [147]. Because the fluorescence quenching ability of GO nanomaterials is very high, it modulates the activity that generates ROS, further expanding the applications of PDT (Figure 5).

Numerous studies have been conducted to achieve tumor targeting, in vivo imaging, and improved PDT effects through functionalization on the GO surface. In one study, PEG-functionalized GO was loaded with the PS 2-(1-hexyloxyethyl)-2-devinyl pyropheophorbide-alpha (HPPH) via supramolecular π-π stacking [148]. HPPH radiolabeled with ^64^Cu enabled in vivo positron emission tomography and fluorescence imaging, resulting in improved cellular uptake of HPPH compared to free HPPH with GO-PEG-HPPH through a more aggressive endocytosis strategy. As a result, GO-PEG-HPPH exhibited enhanced phototoxicity to breast cancer cells when irradiated with light at a wavelength of 671 nm. Through in vivo experiments, mice injected with GO-PEG-HPPH showed a 16-day longer lifespan than mice treated with free HPPH. This indicates that GO-PEG-HPPH utilizing GO as a nanocarrier delivered the drug more efficiently and thereby increased long-term survival. In another study, the PS hypocrelin A (HA) and TiO_2_ nanoparticles were mounted on GO surfaces to form a light-sensitive drug delivery system [149]. By loading TiO_2_ onto GO, ROS could be generated upon exposure to visible light, and the ability to generate ROS was improved through a mutual sensitization mechanism in which a sensitizing effect contributed by the HA-TiO_2_ stable complex. The generated ROS were able to destroy GO, indicating a potential use of this drug delivery system in clinical PDT in terms of metabolism.

In another study, PS Ce6 was conjugated to GO via a redox-responsive cleavable disulfide linker (GO-SS-Ce6) to develop a form that could be released on-demand from cancer cells at significantly higher GSH concentrations compared to normal cells. Therefore, fluorescence and ROS generation were selectively activated by redox agents such as glutathione at high concentrations in tumor cells [150]. On the other hand, in the absence of glutathione, the fluorescence of Ce6 bound to GO was largely quenched due to the FRET process, avoiding the nonspecific excitation and poor targeting ability of PS. The developed GO-SS-Ce6 complex has been proposed as an effective drug delivery vehicle with the strengths of GO’s high surface area and improved chemical tethering properties.

Furthermore, GQDs doped with quantum dots in graphene could provide excellent quantum yield of singlet oxygen as a PDT agent [151]. It is known as a common method to synthesize GQDs using polythiophene as a carbon precursor using hydrothermal methods. The GQDs fabricated in the study were excited by visible light and showed photodynamic activity; their PDT effects were observed through apoptosis of HeLa cells and oncolysis of BALB/nude mice with breast cancer. On the other hand, more advanced studies showed that GQDs could be functionalized and doped with nitrogen and amino groups to show that the amino-N-GQDs exhibited excellent singlet oxygen generation capacity in the NIR region (800 nm) [152].

#### 5.3.4. Upconversion Nanoparticles

Upconversion nanoparticles (UCNPs) are a unique class of optical nanomaterials characterized by their ability to convert low-energy NIR light into high-energy visible/ultraviolet light using a nonlinear anti-Stokes mechanism [153]. The upconversion phenomenon is based on inorganic host crystal lattices doped with trivalent lanthanide ions such as Yb^3+^, Er^3+^, and Tm^3+^. UCNPs require the presence of two different dopant ions [154]. One acts as a sensitizer to absorb NIR radiation, and the other acts as an activator to emit visible light. Two frequently used rare earth ion pairs are ytterbium-thulium (Yb^3+^-Tm^3+^) and ytterbium-erbium (Yb^3+^-Er^3+^). The Yb^3+^ ions act as antennas, absorbing NIR light at about 900–1100 nm and transmitting it to the lanthanide ions, where they mutually upconvert. If this ion is Er^3+^, green and red emission is observed, whereas if it is Tm^3+^, the emitted light is near-ultraviolet, blue and red. In addition, the emission band of UCNP is similar to the band in which PS can be excited, which is characterized by improved ROS production efficiency [128]. In this regard, UCNP may serve as a promising carrier to overcome the limitations of PDT due to the insufficient tissue penetrating ability of short wavelengths (600–850 nm) (Figure 6).

The NaYF_4_: Yb^3+^/ Er^3+^, the first UCNPs used in PDT studies, showed strong emission spectrum in the visible region around 537 and 635 nm when excited by an infrared light source of 974 nm [155]. During the silica coating procedure in the UNCP synthesis, the PS molecule merocyanine 540 (MC-540) was mounted on the nanoparticle. However, the activation wavelength of these PSs is under 700 nm, which is a range in which endogenous molecules such as hemoglobin have strong absorption, a great limitation in their use in PDT. A study successfully detected the generation of singlet oxygen mediated by UCNPs coated with MC-540 with NIR excitation by measuring the decrease in the fluorescence band of the ^1^O_2_ sensor 9,10-anthracenedipropionic acid. Moreover, the first application of UCNP-mediated PDT for in vivo tumor therapy is NaYF_4_:Yb/Er nanoparticles coated with mesoporous silica as nano-transducers and carriers of two different PSs such as MC-540 and ZnPc [156]. Another study found that UCNPs synthesized using dual PS had higher PDT efficacy than using single PS, with improved ROS production capacity and enhanced cytotoxicity. In the tumor-bearing mice, both intratumoral injection of UCNP or intravenous injection of FA and PEG-modified UCNPs (FA-PEG-UCNP) into tumor resulted in tumor growth inhibition at 980 nm excitation. In addition, the tumor-targeting ability and circulating lifespan of UCNP were improved by FA and PEG, respectively, indicating a greater PDT effect when administered intravenously.

One research team prepared NaYF_4_:Er/Yb/Gd upconversion nanocrystals by doping NaYF_4_:Yb/Er UCNP with gadolinium ions and loading them with PS drugs to use as a carrier [157]. Through a water-in-oil inverse microemulsion strategy, methylene blue (MB), a hydrophilic PS drug, was efficiently conjugated to UCNPs in a silica matrix to provide UCNP/MB nanocomposites with a particle size less than 50 nm. The obtained UCNP/MB-based PDT drug successfully generated singlet oxygen at 980 nm excitation, whereas no signal was observed with free MB solution alone or with NaYF_4_:Er/Yb/Gd under the same conditions. Furthermore, polymer-coated NaYF_4_:Yb/Er nanoparticles were used as transport mediators of PS Ce6 to form UCNP-Ce6 supramolecular complexes [158]. Because this UCNP-Ce6 nanosystem showed two emission bands at 550 nm and 660 nm with 980 nm irradiation, PDT performance was improved in that the 660 nm emission wavelength overlapped the absorption band of Ce6, and singlet oxygen production was increased under NIR light irradiation. In particular, there were few observations of UCNPs administered to mice after 1–2 months, demonstrating their nontoxicity to the treated animals.

Although it is common to form NaYF_4_ crystals with a host co-doped with Yb^3+^/Er^3+^ in UCNP-based PDT, doping NaYF_4_ with a Yb^3+^/Tm^3+^ couple shows a similar phenomenon. In one study, NaYF_4_:Yb/Tm UCNPs were coated with a nanometer silica layer, which was further modified with (3-aminopropyl)triethoxysilane APTES using the Stöber method [159]. After that, the UCNPs were covalently bound to PS Ce6 via the amino group of the silica layer. A low concentration (50 μg/mL) of this UCNP-Ce6 nanocomposite was used to kill 50% of CF-7 human breast adenocarcinoma cells at a low dose (7 mW/cm^2^) of 980 nm light for 10 min. Furthermore, they achieved a cell viability greater than 90% under the same conditions without light irradiation, indicating low toxicity of this UCNP-Ce6 nanosystem in the effective concentration range. Alternatively, LiYF_4_:Tm^3+^/Yb^3+^-UCNPs prepared using m-THPC with PS modified with 4-(bromomethyl)benzoic acid performed better when activated with 980 nm NIR irradiation compared to conventional NaYF_4_UCNPs. They emitted an intense blue color and produced a larger amount of singlet oxygen [124,160,161].

More recently, research on multifunctional UCNP-based nanocomposites combining image-guided PDT and multimodal therapy has been attracting attention. UCNPs coated with the PS TiO_2_ were adopted for in vivo PDT image induction that realized complete optical switching in the UV-blue region [162]. In this work, the newly developed photoswitchable upconversion nanoparticles (PUCNPs) were not doped with Nd^3+^. The prominent UV-blue emission of Tm^3+^ characteristic of these PUCNPs was activated at 980 nm excitation and showed excellent photoswitching properties that could be completely deactivated with 800 nm light. As a result, the Tm^3+^ emission band of 350 nm at 980 nm excitation can be well synchronized with the absorption of TiO_2_, which could lead to ROS generation and effective PDT, while also enabling real-time tumor imaging. In contrast, only the Er^3+^ emission band at 660 nm was activated when excited with 800 nm, which made it possible to monitor emission near 650 nm in vivo to track the treatment process.

In addition to the excellent optical properties and abundant surface functions of UCNP, the energy transfer ability of UCNP to deep tissues such as brain tumors efficiently forms ROS, providing a great development for clinical application in the field of brain tumor treatment. However, due to the low upconversion of UCNPs developed with the techniques to date, the quantum yield is less than 3%, and the relatively poor biocompatibility in the physiological environment hinders biological applications. Therefore, more studies on high-efficiency UCNPs considering stability in the future should be conducted.

## 6. Conclusions

The field of PDT has developed rapidly and is constantly being evaluated for new technology. Molecular strategies based on the nanotechnology are being developed to increase the effectiveness and selectivity of PDT. Therefore, numerous organic and inorganic nanoparticles have been newly researched and developed for targeted delivery of photosensitizer pharmaceuticals. This review presents examples of the improved overall effectiveness of PDT cancer treatment by demonstrating that NPs can provide a solution to the important limitations of traditional PS drug delivery. However, because intracranial brain tumors arise from structurally complex and unique organs, such as those surrounded by the blood–brain barrier, compared to other tumors, it is unknown whether they can be completely eradicated using the same approach. Further questions to be explored include whether PDT can be used to treat malignant brain tumors that cannot be resected because of their location. Ongoing study of the various PDTs presented in this manuscript will determine whether advances in cancer research will alleviate morbidity and mortality from treatment of intracranial malignancies and have the potential to revolutionize the treatment of brain tumors. Therefore, while the development of new PDT technology is important, establishment of treatment standards through large-scale clinical practice should be pursued.

## Figures and Tables

**Figure 1 biomedicines-10-00096-f001:**
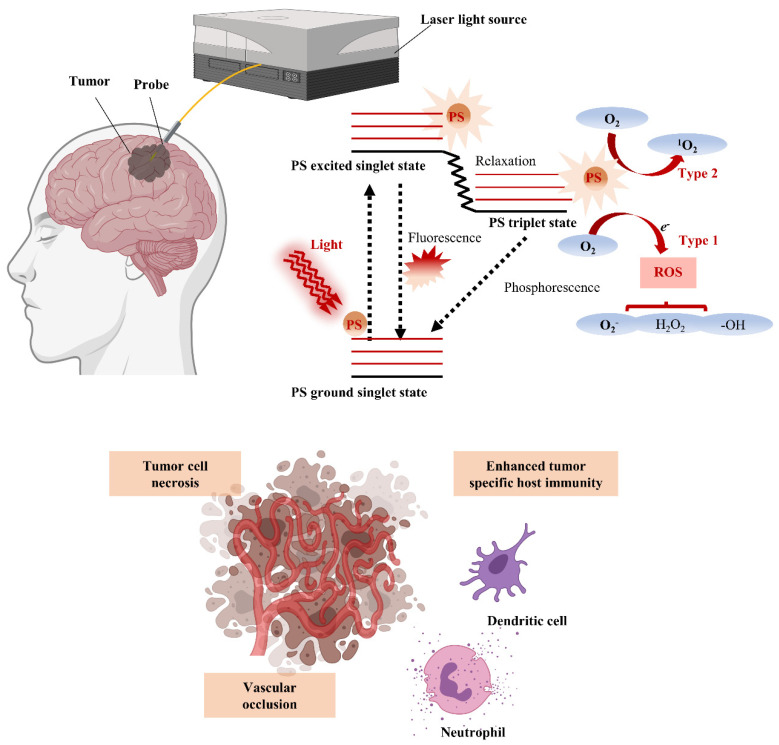
Schematic illustration of photodynamic therapy (PDT) for GBM treatment with energy diagram of the oxygen dependent response. If the photosensitizer (PS) in the ground singlet state is excited by the light wavelength, then the PS in the excited singlet state can convert to the excited triplet state via intersystem crossing. In the presence of molecular oxygen, the PS in the triplet state can undergo a Type 1 or Type 2 redox reaction, producing reactive oxygen species (ROS) that cause tumor cell necrosis, vascular occlusion, and tumor-specific host immunity.

**Figure 2 biomedicines-10-00096-f002:**
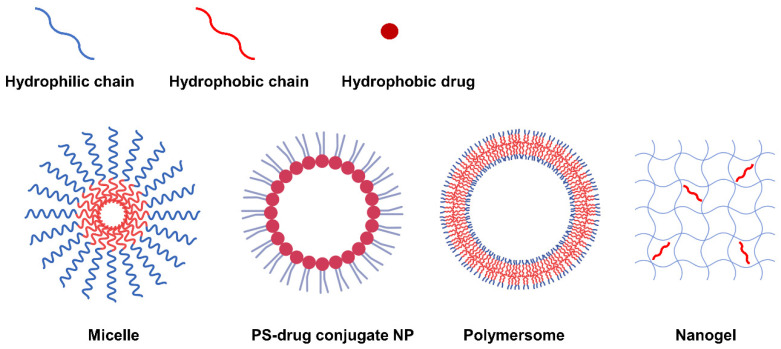
Representative types of NPs classified according to nanostructure.

**Figure 3 biomedicines-10-00096-f003:**
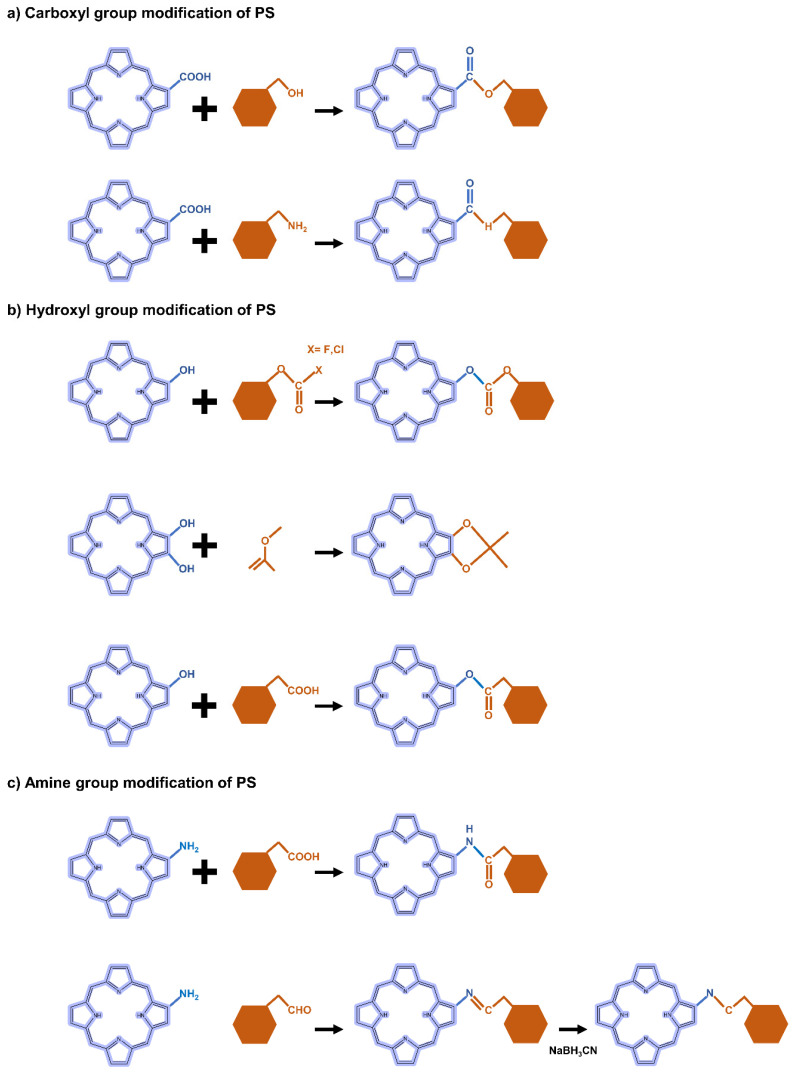
Representative reactions to modify the functional groups of PS-derivative based on (**a**) carboxyl, (**b**) hydroxyl, and (**c**) amine.

**Figure 4 biomedicines-10-00096-f004:**
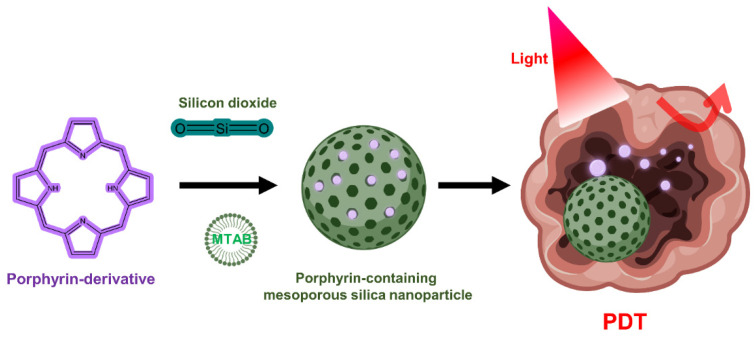
Porphyrin-containing mesoporous silica nanoparticles for PDT.

**Figure 5 biomedicines-10-00096-f005:**
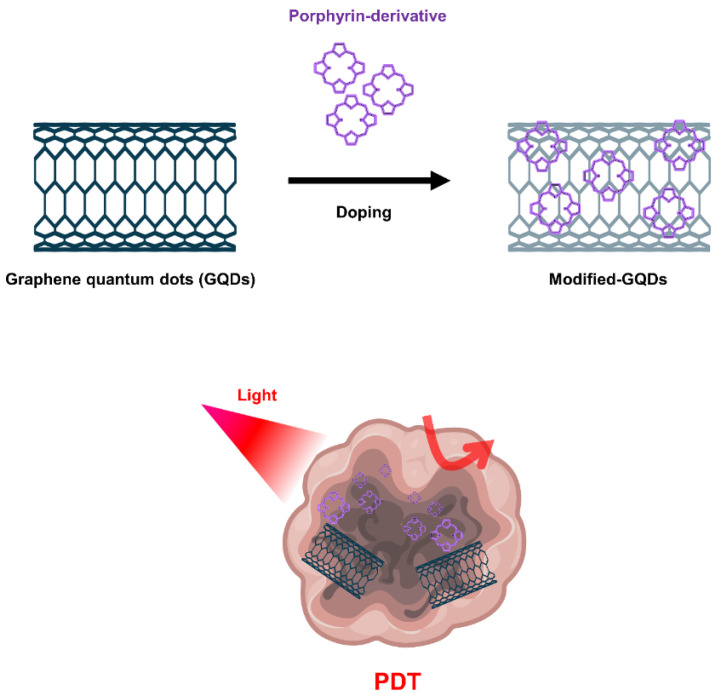
Graphene quantum dots (GQDs)-based nanomaterials for PDT.

**Figure 6 biomedicines-10-00096-f006:**
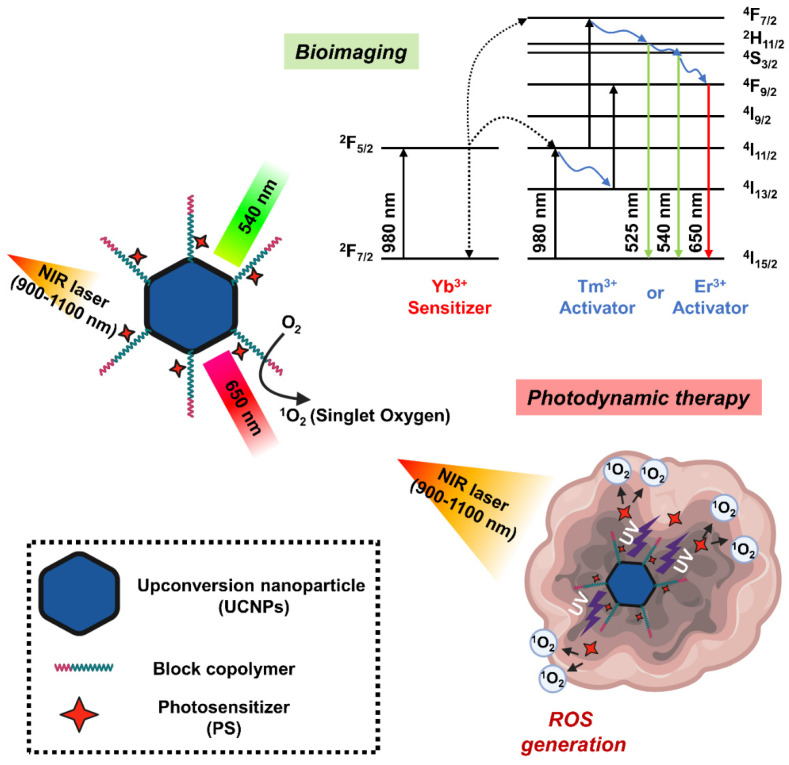
Schematic diagram showing the mechanism of photodynamic therapy and bioimaging through long-wavelength to short-wavelength conversion of upconversion nanoparticles (UCNPs).

**Table 1 biomedicines-10-00096-t001:** WHO classification of brain tumor grades.

	Grade	Tumor Types	Characteristics
Low Grade	Grade I	Craniopharyngioma Chordomas Ganglioglioma Gangliocytoma Pilocytic astrocytoma	Possibly curable via surgery alone Long-term survival Least malignant (benign) Non-infiltrative
	Grade II	Pineocytoma “Diffuse” astrocytoma Pure oligodendroglioma	Slight infiltrative Relatively slow growing Can recur as higher grade
High Grade	Grade III	Anaplastic ependymoma Anaplastic astrocytoma Anaplastic oligodendroglioma	Malignant Infiltrative Tend to recur as higher grade
	Grade IV	Glioblastoma multiforme Medulloblastoma Ependymoblastoma Pineoblastoma	Most malignant Rapidly growing and aggressive Widely infiltrative Recurrence Tendency for necrosis

**Table 2 biomedicines-10-00096-t002:** Recent advances in the preclinical development of nanopharmaceuticals to perform PDT.

Photosensitizer (PS)	Type of Nanomaterials	Tumor Type Treated	Results and Highlights	Year	Ref.
Chlorin e6 (Ce6)	Stem cell membrane-camouflaged bioinspired nanoparticles	Lung cancer	The enhanced antitumor effect of Ng/Ce6@SCV after NIR irradiation significantly inhibits primary tumor growth with fewer side effects.	2020	[80]
	Hyaluronic acid (HA)-based nanomaterials	Primary tumor and melanoma	Multifunctional nanosystem (HPR@CCP) exerted combined photodynamic and immunotherapeutic activity to amplify the therapeutic effect on primary tumors and distant metastases.	2020	[81]
	Peptide p 18-4/chlorin e6 (Ce6)-conjugated polyhedral oligomeric silsesquioxane (PPC) nanoparticles	Breast cancer cells	Cancer-targeting peptide p 18-4/chlorin e6 (Ce6)-conjugated polyhedral oligomeric silsesquioxane (PPC) nanoparticles improved the targeting ability of Ce6 to breast cancer cells to enhance PDT efficacy.	2020	[82]
	Ce6 loaded to the peroxidase-mimic metal-organic framework (MOF) MIL-100 (Ce6@MIL-100)	Breast cancer cell (4T1 cell line)	Peroxidase mimic metal-organic framework efficiently ablated tumors in microenvironment.	2020	[83]
	A fucoidan-based theranostic nanogel consisting of a fucoidan backbone, redox-responsive cleavable linker and Ce6	Human fibrosarcoma cell line (HT1080)	Fucoidan, the polymer backbone of the nanogel platform, enabled cancer targeting by P-selectin binding and enhanced the antitumor effect by inhibiting the binding of vascular endothelial growth factor.	2020	[84]
	Ligation of an anticancer cabazitaxel (CTX) drug via reactive oxygen species-activated thioketal linkage produces a dimeric TKdC prodrug, followed by co-assembly with a photosensitizer, Ce6	Human melanoma patient-derived xenograft (PDX)	Administration of psTKdC NAs followed by laser irradiation produced durable tumor regression, with tumors completely eradicated in three of six PDXs.	2020	[81]
	Light-enhanced PTX nanoparticles (Ce6/PTX2-Azo NPs) were prepared by synthesizing a hypoxia-activated self-sacrificing prodrug of paclitaxel (PTX2-Azo) and encapsulating it with a peptide copolymer decorated with the photosensitizer Ce6	The innately hypoxic microenvironment of most solid tumors	PTX2-Azo prevented premature drug leakage and realized specific release in a hypoxic tumor microenvironment, and the photosensitizer Ce6 efficiently generated singlet oxygen under light irradiation and acted as a positive amplifier to promote the release of PTX	2020	[85]
	Ce6-caspase 3 cleavable peptide (Asp-Glu-Val-Asp, DEVD)-anticancer drug monomethyl auristatin E (MMAE) conjugate, resulting in Ce6-DEVD-MMAE nanoparticles	Squamous cell carcinoma 7 (SCC7)	Light-induced therapeutic strategy based on apoptotic activation of Ce6-DEVD-MMAE nanoparticles can be used to treat solid tumors inaccessible to conventional PDT.	2019	[86]
5-aminolevulinic acid (5-ALA)	Gold nanoparticles (GNP) conjugated to 5-ALA	Nonmelanoma skin cancer Subcutaneous squamous cell carcinoma (cSCC)	GNP conjugated to 5-ALA significantly enhanced the antitumor efficacy of PDT in HaCat and A431 cells	2020	[87]
	Gefitinib PLGA nanoparticles	Lung cancer	Synergistic therapeutic effects were identified by the combination of chemotherapy and photodynamic therapy	2020	[88]
Pheophorbide A (PhA)	Photoactivatable nanomicelles, which are constructed by self-assembly of poly (ethylene glycol) (PEG)-stearamine (C18) conjugate (PTS) with a ROS-sensitive thioketal linker (TL) and co-loaded with doxorubicin (DOX) and photosensitizer pheophorbide A (PhA)	Colon cancer cell line (CT-26)	The gradual elevation of local ROS levels generated by photoactivated PhA synergistically inhibited tumor growth and enhanced anti-tumor immunity by ROS-induced release of DOX.	2020	[89]
	Acid-responsive polygalactose-co-polycinnamaldehyde polyprodrug (PGGA) self-assembled with PhA	Hepatocarcinoma (HepG2)	Intravenous injection of PGCA@PA NPs strongly inhibited tumor growth of hepatocellular carcinoma with negligible side effects.	2020	[90]
	PEG-doxorubicin conjugate	Colon cancer (CT-26)	Synergistically maximized the efficacy of the combination of chemotherapy and photodynamic therapy.	2020	[91]
IR780	IR780 loaded on the prodrug micelle that consisted of camptothecin (CPT) andpolyethylene glycol (PEG) with further modification of iRGD peptide.	Glioma	The targeted prodrug system could effectively cross various barriers to reach the glioma site and greatly enhanced the antitumor effect with laser irradiation.	2020	[92]
	Poly-ε-caprolactone nanoparticles (PCL NPs) modified with LHRH peptide and loaded with IR780 and paclitaxel (PTX)	Ovarian cancer	LHRH peptide modified PCL (PCL-LHRH) NPs demonstrated increased internalization in ovarian tumor cells in vitro and selective targeting in tumor xenografts in vivo	2020	[93]
Indocyanine	Graphene oxide nanoparticle	Osteosarcoma	Nanoparticle consisting of polyethylene glycol (PEG), folic acid (FA), PS indocyanine green (ICG), and doxorubicin inhibited the proliferation and migration of osteosarcoma cells.	2020	[90]
	Self-assembled nanoparticle with indocyanine, camptothecin, RGD peptide	Human cervical carcinoma cell lines (HeLa); Human hepatoma (BEL-7402)	This facile and effective self-assembly strategy to construct nanodrugs demonstrated enhanced performance for cancer theranostics.	2018	[94]

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
