# Peer review of "Nanomedicine in Clinical Photodynamic Therapy for the Treatment of Brain Tumors"

_biomedicines, 2022, doi:10.3390/biomedicines10010096_

Round 1

Reviewer 1 Report

The paper focused on the important problem of PDT applications of nanomedicine for the treatment of glioblastoma multiforme. Authors classified brain tumors according to malignancy and examine the applicability of PDT for each type. I would like to note a detailed review of the types of nanoparticles (NPs), clinical applications of NPs as potential delivery tools for photosensitizer delivery and cancer treatment. The authors mentioned the most important works of the 70s and reviewed in detail the works of the last decade. The question whether PDT can be used to treat glioblastomas that cannot be resected because of their location is discussed. The paper is useful for specialists in the field of nanomedicine and PDT and can be accepted for publication in the journal "Biomedicines".

However, some misprints should be corrected, particularly:

  1. A lot of references are incomplete or with misprints, for example: 4, 8, 9, 17, 26, 29, 33, 72, 74 etc.
  2. Line 450: ‘AuNTs’ should be replaced by ‘AuNRs’.
  3. Line 518: ‘TiO’ should be replaced by ‘TiO2’.

Author Response

Reviewer #1:

The paper focused on the important problem of PDT applications of nanomedicine for the treatment of glioblastoma multiforme. Authors classified brain tumors according to malignancy and examine the applicability of PDT for each type. I would like to note a detailed review of the types of nanoparticles (NPs), clinical applications of NPs as potential delivery tools for photosensitizer delivery and cancer treatment. The authors mentioned the most important works of the 70s and reviewed in detail the works of the last decade. The question whether PDT can be used to treat glioblastomas that cannot be resected because of their location is discussed. The paper is useful for specialists in the field of nanomedicine and PDT and can be accepted for publication in the journal "Biomedicines". However, some misprints should be corrected, particularly:

(Q1) A lot of references are incomplete or with misprints, for example: 4, 8, 9, 17, 26, 29, 33, 72, 74 etc.

Answer: Following your comments, I double-checked all references in this manuscript.

(Q2) Line 450: ‘AuNTs’ should be replaced by ‘AuNRs’. Line 518: ‘TiO’ should be replaced by ‘TiO2’.

Answer: As you mentioned, we carefully double-checked all terminologies in this manuscript.

Reviewer 2 Report

The authors review the use of photodynamic therapy for brain tumor.

The authors started to describe all types of brain tumors, then talked about current clinical trials and finally described various nancarriers for PTD molecules.

In this review there is something important missing, that is the main problem of brain tumor which is the presence of the blood brain barrier.

I this respect, I would have expected the authors to talk about how the drug cross the BBB.

What is the maximum size of a drug or of a drug carrier ? Is this size depend on the type of brain tumor ? Can it be increased (e.g. BBB opening by ultrasounds ). All these questions need to be addressed by the authors !

Author Response

The authors review the use of photodynamic therapy for brain tumor. The authors started to describe all types of brain tumors, then talked about current clinical trials and finally described various nancarriers for PTD molecules.
(Q1) In this review there is something important missing, that is the main problem of brain tumor which is the presence of the blood brain barrier. In this respect, I would have expected the authors to talk about how the drug cross the BBB. What is the maximum size of a drug or of a drug carrier? Is this size depend on the type of brain tumor? Can it be increased (e.g. BBB opening by ultrasounds). All these questions need to be addressed by the authors !
Answer: Based on your comments, we carefully modified the Introduction section, and newly added section #3 to reflect your comment. Before addition in the Introduction section Clinical trials for brain tumors exist to date, but there are many questions about PDT and its usefulness as a standard adjuvant therapy. First, the effect of PDT alone reported in clinical trials were positive based on parallel administration of standard treatment. Vari-ables that should be standardized across further studies include photosensitizer (PS) se-lection, injected dose, irradiation light wavelength, sensitivity of brain tumor types, and adjuvant use of chemotherapy and radiation. Furthermore, additional studies are needed to enhance the targeting of brain tumors while considering the pharmacokinetic aspects and methods of improving the quantum yield of the PS, which generates effective reactive oxygen species under light irradiation. In this regard, the rapidly developing fields of nanotechnology and nanomedicine are producing nanostructured materials that can overcome the shortcomings of
delivery systems used in clinical practice. Nanoparticles can increase the low solubility of a PS, prolong its blood circulation, and promote targeted delivery and cellular uptake, while protecting the drug from degradation. This makes it an interesting alternative to traditional PDT as the nanostructures can enable efficient transport of PS and ameliorate the lack of anticancer activity [8]. To date, in addition to micellar self-assembly techniques for PS delivery, numerous nanoparticles, such as gold, silica, upconversion, and carbon-based particles, have been studied to increase their phototoxic properties and to increase their
concentrations in tumor sites.
In this study, we classify brain tumors according to malignancy and examine the
applicability of PDT for each type. We discuss the use of PDT and the properties and clinical applications of nanoparticles as potential delivery tools for PS delivery. In addition, the possibility of application to brain tumors is discussed through clinical cases of nanomedicine-based PDT.
After addition in the Introduction section
Clinical trials for brain tumors exist to date, but there are many questions about PDT and its usefulness as a standard adjuvant therapy. First, the effect of PDT alone reported in clinical trials were positive based on parallel administration of standard treatment. Vari-ables that should be standardized across further studies include photosensitizer (PS) se-lection, injected dose, irradiation light wavelength, sensitivity of brain tumor types, and adjuvant use of chemotherapy and radiation. Furthermore, additional studies are needed to enhance the 4
targeting of brain tumors while considering the pharmacokinetic aspects and methods of improving the quantum yield of the PS, which generates effective reactive oxygen species under light irradiation. In this regard, the rapidly developing fields of nanotechnology and nanomedicine are producing nanostructured materials that can overcome the shortcomings of
delivery systems used in clinical practice. In fact, the functional presence of the blood-brain barrier (BBB) limits the delivery of drugs to the brain tumors. To overcome this limitation, many strategies for temporarily opening the BBB through physical impact such as magnetic resonance (MR)-guided focused ultrasound have been studied recently, but this raises a problem in compatibility [8]. Therefore, the use of multifunctional nanocarriers as drug delivery systems is emerging as one of the most promising strategies [9]. In general, intracellular transport of nanocarriers is mediated by the vesicular system, and three types of
intracellular vesicles are involved (e.g. clathrin-mediated, caveo-lae-mediated and macropinocytotic vesicles) [10]. Therefore, in order to pass through these pathways, nanocarriers covalently bound with specific targeting ligands to guide the drug across the BBB to specific sites in each tumor type. The physicochemical and mechanical properties of nanocarriers differ depending on the material, size, shape (mesoporous structure, rod shape, particle), and the selected ligand. This allows customization for in-creased brain-targeted
delivery of PS or therapeutic drugs. Although many PS nanocarriers are still in the early stages of translation, many advances have been made in recent years for functional nanomedicines based on BBB crossing.
Another advantage of nanoparticles is that they can increase the low solubility of PS, prolong blood circulation, promote targeted delivery and cellular uptake, while protecting the drug from degradation. This makes it an interesting alternative to traditional PDT as the nanostructures can enable efficient transport of PS and ameliorate the lack of anticancer activity [11]. To date, in addition to micellar self-assembly techniques for PS delivery, numerous nanoparticles, such as gold, silica, upconversion, and carbon-based particles, have been studied to in-crease their phototoxic properties and to increase their concentrations in
tumor sites.
In this study, we classify brain tumors according to malignancy and examine the
applicability of PDT for each type. We discuss the use of PDT and the properties and clinical applications of nanoparticles as potential delivery tools for PS delivery. In addition, the possibility of application to brain tumors is discussed through clinical cases of na-nomedicine-based PDT.
Newly organized section 3
3. Strategies to improve permeability of nanocarrier through the blood-brain barrier
One of the major limitations of treating brain tumors is the difficulty of delivering drugs to the brain. The brain is surrounded by the blood-brain barrier (BBB), a selective barrier formed by endothelial cells in the cerebral microvessels, which regulates nutrient and ion transport and protects the brain from neurotoxic molecules to maintain brain homeostasis[34]. Unfortunately, most drugs cannot cross the BBB via physiological pathways due to the extreme selectivity of the barrier, which constitutes the greatest obstacle to systemic treatment for most central nervous system (CNS) diseases. In the recent decade, many strategies have been studied, such as topical delivery, implantation of a sustained drug-release scaffold [35], nasal administration [36], ultrasound to tem-porarily open the BBB [37], and nanoparticle functionalization to enhance BBB penetration [38]. However, local drug delivery methods are considered highly invasive as they require procedural surgery. In addition, the intranasal route has a disadvantage in that the delivered dose varies greatly depending on the condition of the nasal mucosa. Therefore, despite the difficulties across the
BBB, the most popular and well-studied de-livery route remains the systemic route through the functionalization of nanoparticles.
Nanocarriers can traverse the BBB using a variety of physiological pathways, including receptor-mediated transcytosis (RMT) or adsorption-mediated transcytosis (AMT). To achieve this goal, many nanocarrier systems, such as inorganic, polymeric, or li-pid-based nanoparticles, have been developed and shown to cross the BBB due to their tailored surface properties. Numerous studies have demonstrated that physically coating nanoparticles with surfactants and chemical functionalization with specific ligands is a successful strategy to
enhance BBB traverse via the physiological pathways mentioned above [39,40]. The size and charge of nanoparticles are also aspects that can affect brain penetration, but if the surface functionalization is done properly, there is no significant difference in a wide size range (from 5 to 400 nm) [41]. Smaller nanoparticles can cross the BBB more easily and diffuse better through the brain, but larger nanoparticles can also cross the BBB in slightly smaller amounts when properly functionalized. On the other hand, larger particles can load a greater
amount of drug but reach the brain at a lower concentration, and smaller nanoparticles cannot contain a large amount of drug but reach the brain at a higher concentration. Therefore, the key to increasing the amount of drug delivered to the brain is finding the optimal particle size and designing a nanoparticle system that fits the purpose.

New Addition in Reference sectioin

8. Zhang, Y.H.; Wang, B.L.; Zhao, R.B.; Zhang, Q.; Kong, X.D. Multifunctional
nanoparticles as photosensitizer delivery car-riers for enhanced photodynamic cancer therapy. Mat Sci Eng C-Mater 2020, 115.
9. Biratu, E.S.; Schwenker, F.; Ayano, Y.M.; Debelee, T.G. A survey of brain tumor
segmentation and classification algorithms. J Imaging 2021, 7.
10. Khan, M.A.; Lali, I.U.; Rehman, A.; Ishaq, M.; Sharif, M.; Saba, T.; Zahoor, S.;
Akram, T. Brain tumor detection and classifi-cation: A framework of marker-based watershed algorithm and multilevel priority features selection. Microsc Res Tech 2019, 82, 909-922.
34. Achar, A.; Myers, R.; Ghosh, C. Drug Delivery Challenges in Brain Disorders across the Blood-Brain Barrier: Novel Methods and Future Considerations for Improved Therapy. Biomedicines 2021, 9, doi:10.3390/biomedicines9121834.
35. Chaichana, K.L.; Pinheiro, L.; Brem, H. Delivery of local therapeutics to the brain: working toward advancing treatment for malignant gliomas. Ther Deliv 2015, 6, 353-369, doi:10.4155/tde.14.114.
36. Lochhead, J.J.; Thorne, R.G. Intranasal delivery of biologics to the central nervous system. Adv Drug Deliv Rev 2012, 64, 614-628, doi:10.1016/j.addr.2011.11.002.
37. Abbasi, J. Guided Ultrasound Opens Blood-Brain Barrier to Cancer Drugs. JAMA 2021, 326, 1785, doi:10.1001/jama.2021.19638.
38. Lombardo, S.M.; Schneider, M.; Tureli, A.E.; Gunday Tureli, N. Key for crossing
the BBB with nanoparticles: the rational design. Beilstein J Nanotechnol 2020, 11, 866-883, doi:10.3762/bjnano.11.72.
39. Khongkow, M.; Yata, T.; Boonrungsiman, S.; Ruktanonchai, U.R.; Graham, D.;
Namdeel, K. Surface modification of gold nanoparticles with neuron-targeted exosome for enhanced blood-brain barrier penetration. Sci Rep-Uk 2019, 9, doi:10.1038/s41598-019-44569-6.6
40. Del Amo, L.; Cano, A.; Ettcheto, M.; Souto, E.B.; Espina, M.; Camins, A.; Garcia,
M.L.; Sanchez-Lopez, E. Surface Functionalization of PLGA Nanoparticles to Increase Transport across the BBB for Alzheimer's Disease. Appl Sci-Basel 2021, 11, doi:10.3390/app11094305.
41. Jo, D.H.; Kim, J.H.; Lee, T.G.; Kim, J.H. Size, surface charge, and shape determine therapeutic effects of nanoparticles on brain and retinal diseases. Nanomedicine-Uk 2015, 11, 1603-1611, doi:10.1016/j.nano.2015.04.015.

Round 2

Reviewer 2 Report

no further comments !